# Comparing non-breeding distribution and behavior of red-legged kittiwakes from two geographically distant colonies

**Brie A. Drummond**[1]*, **Rachael A. Orben**[2], **Aaron M. Christ**[1], **Abram B. Fleishman**[3], **Heather M. Renner**[1], **Nora A. Rojek**[1], **Marc D. Romano**[1]

**1** Alaska Maritime National Wildlife Refuge, U.S. Fish and Wildlife Service, Homer, Alaska, United States of America, **2** Department of Fisheries and Wildlife, Oregon State University, Corvallis, Oregon, United States of America, **3** Conservation Metrics, Inc., Santa Cruz, California, United States of America

☯ These authors contributed equally to this work.
* Brie_Drummond@fws.gov

**Data Availability Statement:** Data are available on ServCat (Buldir data; https://doi.org/10.7944/

## Abstract

Knowledge of non-breeding distributions is a crucial component to seabird conservation, as conditions during the non-breeding period can play an important role in regulating seabird populations. Specifically, if seabirds from different colonies share the same wintering grounds, conditions in that shared region could have a widespread impact on multiple breeding populations. Red-legged kittiwakes (*Rissa brevirostris*) are endemic to the Bering Sea and may be especially susceptible to effects of climate change due to a restricted breeding range, small population size, and specialized diet. To examine whether red-legged kittiwakes from different breeding colonies overlapped in winter distribution and activity patterns, we used geolocation loggers to simultaneously track individuals from the two largest red-legged kittiwake breeding colonies in Alaska (separated by over 1000 km) during two consecutive non-breeding periods. We found that non-breeding activity patterns were generally similar between birds originating from the two colonies, but birds employed different migratory strategies during the early winter. Kittiwakes from Buldir Island in the western Aleutian Islands left the colony in September and immediately headed west, spending October through December around the Sea of Okhotsk and the Kuril Islands. In contrast, birds from St. George Island in the Pribilof Islands remained in the eastern Bering Sea or around the eastern Aleutian Islands for a couple months before traveling farther west. During late winter however, from January through March, birds from both colonies converged south of Kamchatka and east of the Kuril Islands over the Kuril-Kamchatka Trench and in the Western Subarctic Gyre before returning to their respective colonies in the spring. This late winter overlap in distributions along the Kuril-Kamchatka Trench suggests the region is a winter hotspot for red-legged kittiwakes and highlights the importance of this region for the global kittiwake population.

P9SC9A70) and Dryad (St. George data; https://doi.
org/10.5061/dryad.xksn02vgb).

**Funding:** The North Pacific Research Board (www.
nprb.org) provided funding for RAO and AMF
(grant #1409 to RAO). The United States Fish and
Wildlife Service provided funding for BAD, AMC,
HMR, MDR, and NAR. Conservation Metrics, Inc.
employs AMF but did not provide any salary or
other financial support for this project. The funders
played no role in study design, data collection and
analysis, decision to publish, or preparation of the
manuscript.

**Competing interests:** The authors have read the
journal's policy and the authors of this manuscript
have the following competing interests: Abram
Fleishman is employed by the commercial
company Conservation Metrics, Inc. This does not
alter the authors' adherence to PLOS ONE policies
on sharing data and materials.

## Introduction

Understanding the factors that influence the regulation of populations is critical to species conservation. During the non-breeding period, seabirds disperse from breeding colonies and may travel tens of thousands of kilometers across the ocean [1, 2]. Conditions seabirds experience during this time may affect not only overwinter survival [3] but may also have carryover effects that influence subsequent breeding season productivity and survival [4–6]. Indeed, extrinsic factors during the non-breeding months can have marked effects on the demography of many seabird populations [7–10]. Major threats to seabirds when at sea include decreased food availability due to climate change or overfishing, fisheries bycatch, pollution, and severe storms [11–18]. Knowing where seabirds spend the non-breeding period is therefore necessary in assessing and managing risk in those populations.

Diversity of habitat use can influence a population's vulnerability to environmental change [19]. Widely distributed habitat generalists may be more resilient to environmental change than more narrowly distributed habitat specialists [20–22]. Likewise, "hotspots" that consistently concentrate large numbers of seabirds during the non-breeding period due to favorable foraging conditions are especially important areas for seabird conservation [23–25]. For example, during the winters of 2007 to 2010, an estimated 85% of the eastern Canadian breeding population of common murres congregated together in areas that also had high risk of disturbance and pollution from oil exploration [26]. When birds from separate colonies share wintering regions, increased threats or decreased food availability in those areas may lead to widespread population declines [27–29].

Red-legged kittiwakes (*Rissa brevirostris*) are colonial, cliff-nesting seabirds endemic to the Bering Sea, with just a few remote breeding locations in Alaska and Russia [30]. With a restricted breeding range, small population size (ca 280,000 individuals [31]), and specialized diet relying heavily on myctophid fish in the summer [32–34], red-legged kittiwakes may be especially vulnerable to a rapidly changing climate [35]. Population trends, patterns in annual breeding success, and diets of red-legged kittiwakes vary by colony [34, 36], Commander Islands Nature and Biosphere Reserve unpubl. data]. Non-breeding distributions have been studied only for red-legged kittiwakes from St. George Island in the Pribilof Islands [37–39], the world's largest colony with about 235,000 birds (>80% of the global population [40]). Knowledge of whether red-legged kittiwakes from other colonies share wintering grounds would provide important information for the conservation of the global population.

We explored the extent to which red-legged kittiwakes from the two largest breeding colonies in Alaska overlapped in winter distribution and activity patterns. We used geolocation loggers to simultaneously track red-legged kittiwakes from St. George Island in the Pribilof Islands and Buldir Island (about 9,000 birds [40]) in the western Aleutian Islands over two consecutive non-breeding periods in 2016–17 and 2017–18. Given that previous tracking studies have shown red-legged kittiwakes from St. George travel past Buldir Island to get to their wintering grounds and show population level site fidelity to their wintering region across multiple years [37–39, 41], we predicted that individuals from St. George and Buldir would overlap in winter locations. Likewise, we also predicted that winter location would influence activity patterns regardless of colony of origin, and thus the amount of time birds spent foraging, flying, and resting on the water would be similar between individuals from both colonies.

## Materials and methods

### Ethics statement

All bird handling was approved by the Institute of Animal Care and Use Committee (IACUC) of the U.S. Fish and Wildlife Service (USFWS) Alaska Region and Oregon State University. Permits were provided by USFWS and the State of Alaska.

## Field methods

We used geolocation loggers (C-65, Migrate Technology) to record light level and saltwater immersion data from red-legged kittiwakes at Buldir Island (52° 21' N, 175° 56' E; n = 5 in 2016–17, n = 8 in 2017–18) and St. George Island (56° 36' N, 169° 33' W; n = 29 in 2016–17, n = 7 in 2017–18). Both colonies are long-term seabird monitoring sites of the Alaska Maritime National Wildlife Refuge, where kittiwake productivity, population, survival, and diet data have been collected for several decades. Geolocation loggers sampled light levels (clipped at a maximum of 1000 lux) every minute, and recorded the maximum light value every 5 minutes. Saltwater immersion (based on a conductivity score of > 63) was measured every 6 seconds, with the number of wet samples tallied in 5-minute blocks.

Birds were captured at their nest sites during the breeding period (June and July) using noose poles and foot snares from above and below the nests. All birds were banded with a uniquely numbered metal band on one leg and a plastic band with a geolocator attached to the band with zip ties on the other leg. In successive years, birds were recaptured and loggers and plastic bands were removed; no birds were retagged upon retrieval of loggers. Feather or blood samples were collected at logger deployment or recapture for molecular sex determination (Avian Genetics Inc., Tallahassee FL) from all Buldir birds and St. George birds captured in 2016. Loggers were 14x8x6mm in size; the weight of the logger and band set-up combined was approximately1.5g, which is <0.5% of the mass of any bird captured (316-440g) and well below the accepted standard for avian geolocators of 3% proportion of body mass [42, 43]. While we cannot discount effects of the devices on movement and foraging [44], deleterious effects of loggers are likely less pronounced in species with low wing loading like kittiwakes [45].

Estimates of colony-level breeding success were calculated by monitoring a subsample of visible kittiwake breeding sites (Buldir: n = 39 in 2016, n = 41 in 2017; St. George: n = 231 in 2016, n = 153 in 2017) on long-term monitoring plots. Throughout the breeding period (late May through August), nest contents were observed at 3–7 day intervals using binoculars or spotting scopes [46, 47]. Plots followed for breeding success estimates were not used for geolocator capture.

## Analysis

**Non-breeding distributions.** Errors around the location data make it difficult to tell exactly when birds leave the colony (especially in individual birds that do not immediately make long post-breeding period migrations). We considered September through April the non-breeding period, based on timing of breeding from Buldir and St. George, where eggs are laid in late May or early June and chicks fledge in August [48, 49]. Although some birds undoubtedly were still at the colony during early September and arrived back sometime during April, this time window captures the extent of the non-breeding migration for all birds.

Light level data from the geolocators were analyzed following methodology described in a previous red-legged kittiwake geolocator study [39]. We determined daily twilights using twGeos 0.1.2 (github.com/slisovski/twGeos [50, 51]), with a light threshold of 0.5 and sun angle of -7 to -1. We then estimated highest probable twice-daily locations and the resulting most-probable tracks using probGLS which has a median error of up to 185km at similar latitudes [52]. Because the probGLS package does fairly well in minimizing error around the fall and spring equinoxes (median error of 145km [52]), and visual examination of the twice-daily locations within a week on either side of the equinoxes did not show any obvious bias, we did not exclude any data points around the equinox periods. We calculated utilization distributions (UDs) using the kernelUD function in the adehabitatHR package [53]. To quantify

overlap in core distributions (50% UD) and overall range (95% UD) between colonies, we calculated Bhattacharyya's affinity (BA), in which BA values can vary from 0 (no overlap) to 0.50 (complete overlap) for 50% UD and 0 (no overlap) to 0.95 (complete overlap) for 95% UD [39, 54]. When months were combined for any figures or analyses, we excluded any birds without full datasets for the whole period of interest.

We examined the relative dispersal of birds by comparing the average daily distances between birds over the non-breeding period. Using the midpoint of twice-daily location estimates, we calculated average distance between all pairs of birds each day between September 1 and April 30. To determine if birds from one colony were closer to each other than to those from the other colony, we calculated the difference between the average between-colony distances and within-colony distances for each day. A difference of 0 on a given day indicates that birds were equally close to other birds from both colonies; increasing positive values indicate greater spatial separation between colonies, whereas negative values indicate birds were closer to those from the other colony than their own. Bootstrap confidence intervals were calculated for both daily and monthly average distances (1000 resamples), but it should be noted that they likely underrepresented the actual variability due to the small sample sizes.

Finally, we calculated the distance from the colony and cumulative distance traveled between September 1 and April 30, based on twice-daily locations, using only birds with a complete data series during that time period. We examined whether maximum distance from the colony and cumulative distance traveled during the non-breeding period varied as a function of year or island using an ANOVA with the lm function in the 'stats' package of R [55]. Simultaneous tests for comparisons within these models were calculated using the glht function in the mucltcomp package version 1.4–15 [56].

**Activity patterns.** We used saltwater immersion data to classify kittiwake activity into three behavior categories [37, 57, 58]: (1) on water, when a logger was wet for $\geq$ 98% of a 5-minute period (allowing for occasional dry records due to bathing or preening), (2) dry, when a logger was dry for an entire 5 minute period, likely indicative of flight, although some birds may be occasionally roosting (e.g., on boats, floating debris), and (3) active foraging, when a logger had alternating wet and dry readings in a 5-minute period. As surface feeders, kittiwakes primarily catch prey by using short plunge dives from the air [33, 57, 59], which likely cause alternating wet and dry measurements due to active splashing. This likely underestimates actual foraging time, as "sit-and-wait"-type foraging when a bird is floating on the surface and dipping only its' beak into the water would be missed. Wet/dry datasets were restricted to October through March as some birds could have been at the colony during September and April, which could have resulted in different behavior (i.e., roosting and attending nest sites on land). Only birds with a complete wet/dry data series during October through March were included.

As the three behavior classes were related (i.e., a bird that spends more time in flight will have less time for other activities), we used James second-order tests [60] with nonparametric p-values estimated with 1000 bootstrap resamples to compare the proportion of time birds spent on the water, flying, and foraging between years and colonies. To examine behavior during different periods of the day, we used light level data recorded by the loggers to classify activities as occurring during daytime, nighttime, and in 1-hour windows immediately surrounding sunrise and sunset (as determined by twGeos [50]).

**Breeding success.** To quantify breeding success at the colonies, we estimated several measures of kittiwake success at each colony each year. Red-legged kittiwakes lay a single egg a season. From counts of nesting attempts, nests with eggs, and nests with chicks hatched and fledged, we calculated total proportions of laying success (nests with eggs laid/nests), hatching success (nests with chicks hatched/nests with eggs laid), fledging success (nests with chicks

fledged/nests with chicks hatched), and overall breeding success (nests with chicks fledged/ nests) [47].

All data processing, analysis, and statistical tests were performed in R version 4.0.3 [55], with significance of p<0.05. All maps were created using the open source mapdata package 2.3.0 [61].

## Results

### Non-breeding distribution

In both years of our study, red-legged kittiwakes from the two colonies spent the early winter months in different areas of the northern Pacific Ocean (Fig 1). Most Buldir birds left the colony in September and immediately headed west towards the Kuril Islands, spending October and November around the Kuril Islands and in the Sea of Okhotsk, and December east of the Kuril Islands over the Kuril-Kamchatka Trench (with the exception of one bird that traveled north into the Bering Sea in November 2017). In contrast, birds breeding at St. George mostly remained in the Bering Sea around the Pribilof Islands (2016–17), around the eastern Aleutian Islands (2017–18), and along the eastern coast of Kamchatka (both years). Average distance between birds from the different colonies, after correcting for within-colony dispersion, was at least 1000km during October and November in both years, and more than 500km in December (Fig 2). BA values for 50% UDs between colonies from October through December were close to zero ($BA_{50} = 0.010 \pm 0.059$ in 2016–17; $BA_{50} = 0.000 \pm 0.056$ in 2017–18), indicating very little overlap in core distribution during early winter (Fig 1). Overall range (95% UD) during the early winter did overlap to a small degree ($BA_{95} = 0.276 \pm 0.058$ in 2016–17; $BA_{95} = 0.182 \pm 0.072$ in 2017–18).

During the late winter, birds from both Buldir and St. George were concentrated south of Kamchatka and east of the Kuril Islands over the Kuril-Kamchatka Trench in January, began to move back east towards the Aleutian Island chain in February or March, and by April were back at their respective breeding colonies. Overlap in distributions between the colonies during the late winter was higher compared to the early winter in both years (Fig 3), demonstrated in both core distribution ($BA_{50} = 0.178 \pm 0.059$ in 2016–17, $BA_{50} = 0.335 \pm 0.033$ in 2017–18) and overall range ($BA_{95} = 0.605 \pm 0.057$ in 2016–17, $BA_{95} = 0.776 \pm 0.032$ in 2017–18). There was little difference in distances within and between colonies from January through March, indicating they were as close to birds from the other colony as to their own (Fig 2), with the exception of January 2017 when St. George birds were slightly segregated from Buldir birds. In 2016–17, only 61% (n = 17) of the St. George birds traveled further south and east to overlap with core UD of the Buldir birds ($BA_{50} = 0.229 \pm 0.045$), whereas 39% of the St. George birds (n = 11) stayed along the Kamchatka Peninsula and the northern Kuril Islands, with no overlap in the core distribution with the birds from Buldir ($BA_{50} = 0.000 \pm 0.051$; Fig 4). These different St. George groups had little overlap with each other ($BA_{50} = 0.026 \pm 0.047$; Fig 4) and were not significantly segregated by sex ($X^2_{1, 28} = 2.80$, p = 0.094).

From September through April, birds from Buldir covered significantly more cumulative distance than birds from St. George in 2016–17 (t = 5.956, p<0.001; Table 1) but not in 2017–18 (t = -0.522, p = 0.940; Table 1). Birds from St. George traveled a farther maximum distance from the colony compared to birds from Buldir only in 2017–18 (t = -2.646, p = 0.044; Table 1).

### Activity patterns

Kittiwakes from both colonies spent 69.7 ± 3.2% of the time from October through March on the water; the remaining time was spent flying (18.3 ± 2.6%) and foraging (12.0 ± 1.6%). There

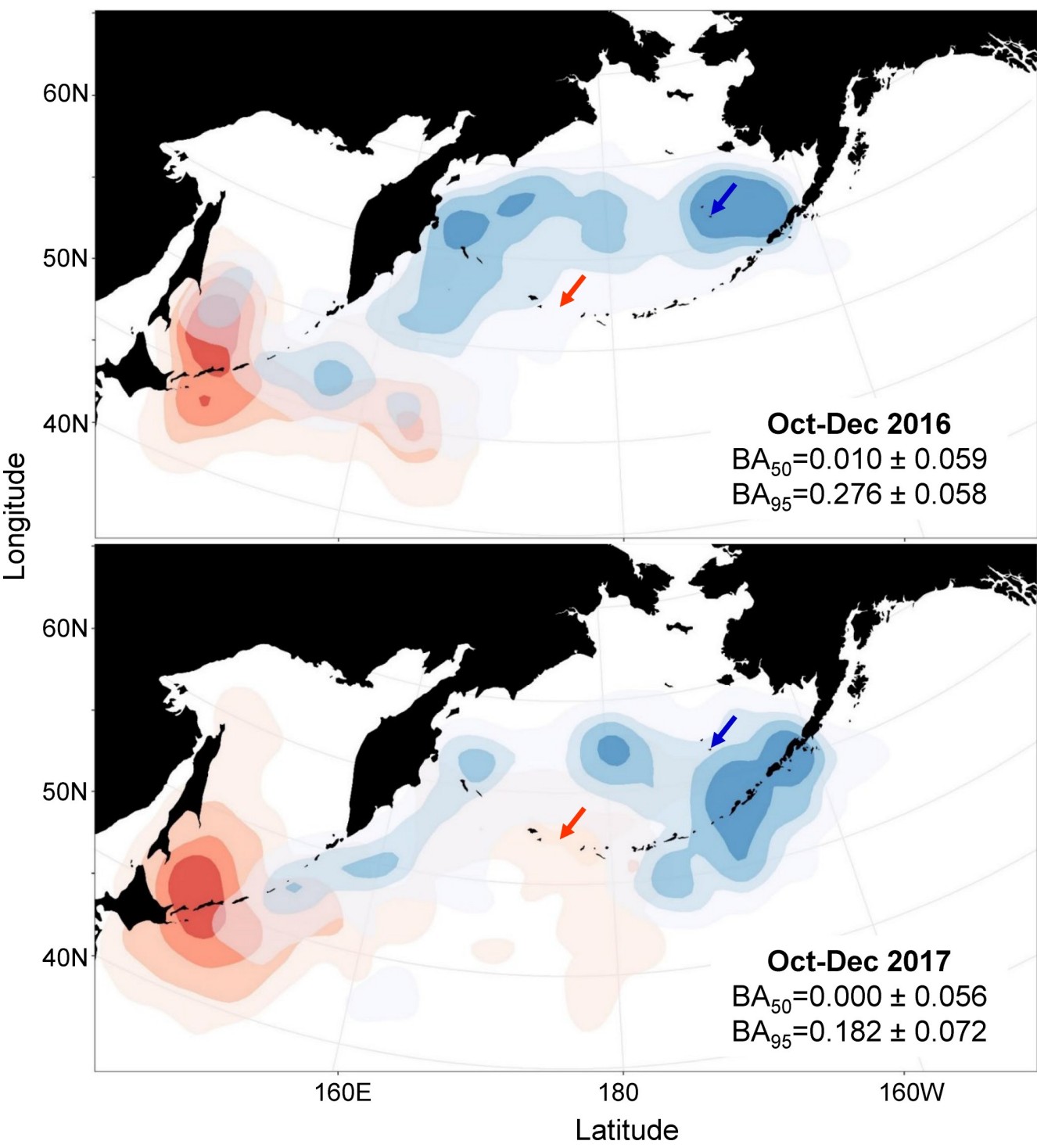

**Fig 1. At-sea utilization distributions (UD) of red-legged kittiwakes in October-December 2016 and 2017.** Birds from Buldir Island (n = 5 in 2016, n = 9 in 2017) are shown in red, birds from St. George Island (n = 29 in 2016, n = 7 in 2017) in blue; arrows indicate breeding colony locations. The 25%, 50%, 75%, and 95% UD levels are shown in progressively lighter shades. BA values represent amount of overlap between each colony in core distribution (50% UD, out of a maximum of 0.5) and overall range (95% UD, out of a maximum of 0.95).

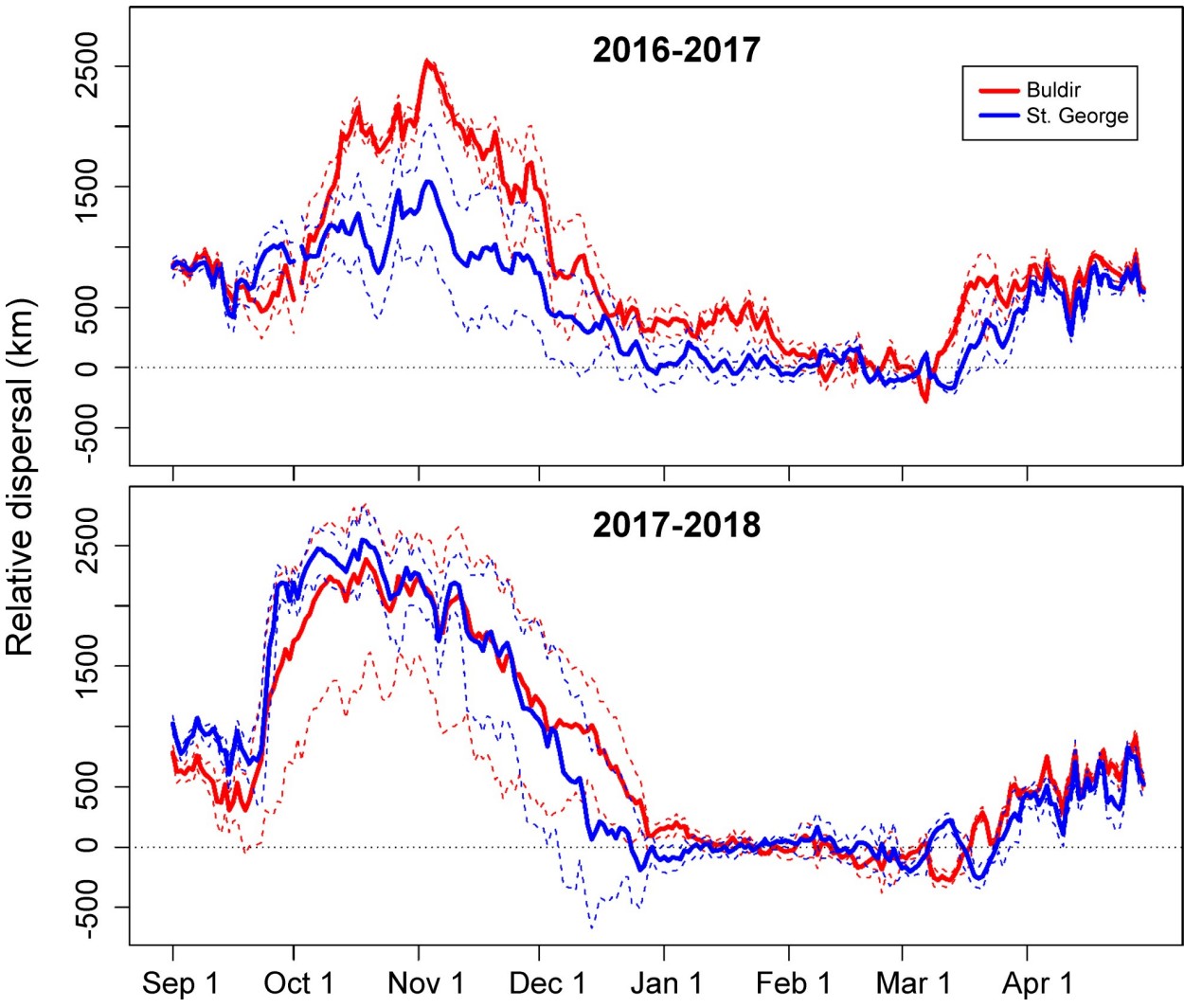

**Fig 2. Dispersal of red-legged kittiwakes from Buldir and St. George islands during winters 2016–2017 and 2017–2018.** Disperal is calculated as the mean daily difference in between-colony and within-colony distances (km). A zero value indicates birds were equally close to other birds from both colonies; increasing positive values indicate greater spatial segregation between colony, whereas negative values indicate birds were closer to those from the other colony than their own. Dashed lines represent confidence intervals calculated by bootstrapping.

was no significant difference in the amount of time kittiwakes from Buldir spent on the water, flying, and foraging between years (Table 2, p = 0.245). Birds from St. George exhibited a small but significant difference in activity between years (p = 0.014), with slightly more time flying and less time on the water in 2016–17 compared to 2017–18 (Table 2), despite traveling less cumulative distance in 2016–17.

Between colonies, activity patterns did not differ significantly over the whole non-breeding period (Table 2, p = 0.091) or the early winter months from October through December (Table 2, p = 0.114). During the late winter period from January through March, birds from the different colonies showed small but significant differences in activity (p = 0.002), with St. George birds spending more time on the water and less time flying than Buldir birds (Table 2).

Behavior varied by time of day in a similar way for birds from both colonies (Fig 5). From sunset until an hour before sunrise, birds were on the water 94.8 ± 3.2% of the time, with very

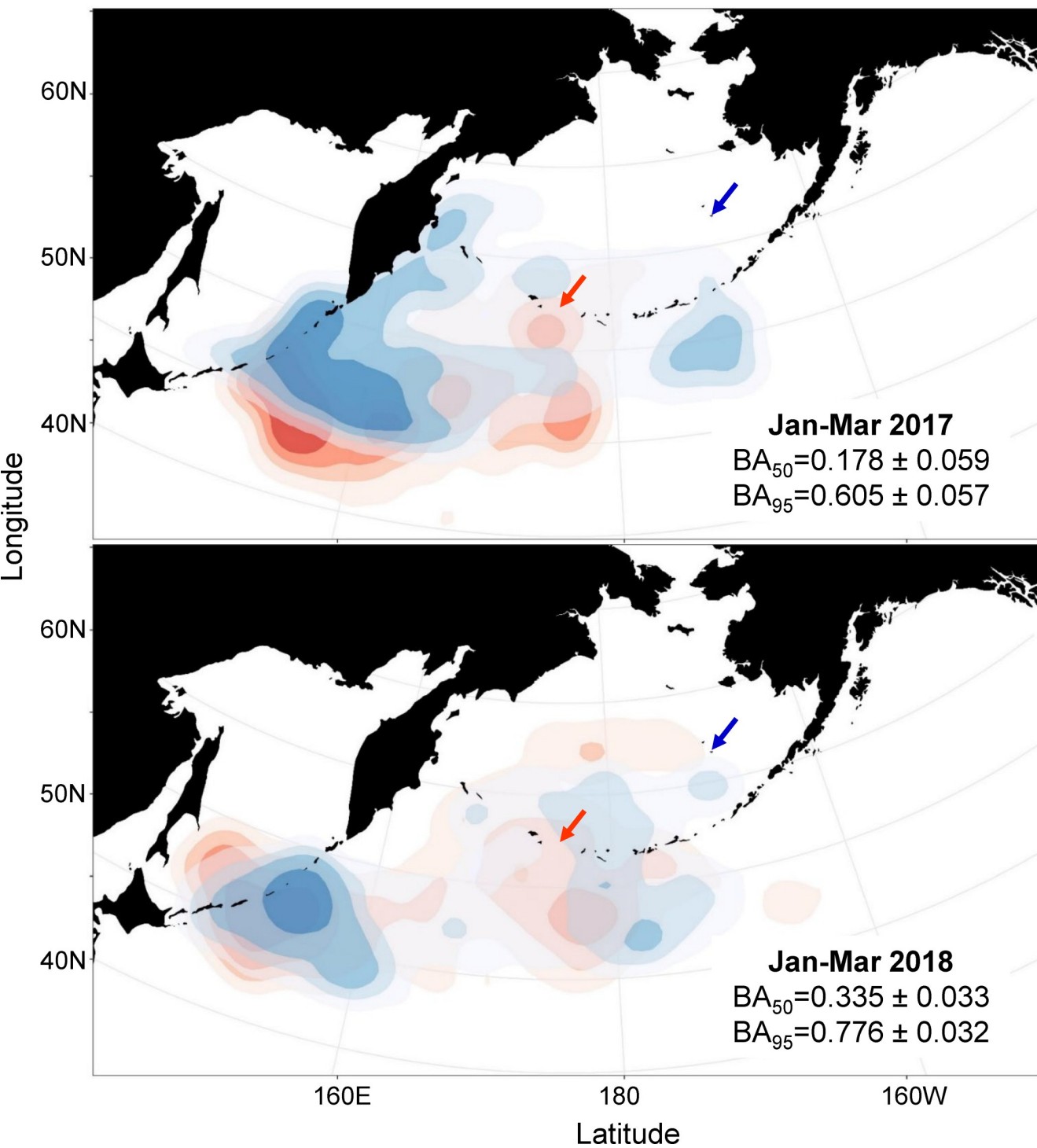

**Fig 3. At-sea utilization distributions (UD) of red-legged kittiwakes in January-March 2017 and 2018.** Birds from Buldir Island (n = 5 in 2017, n = 8 in 2018) are shown in red, birds from St. George Island (n = 28 in 2017, n = 6 in 2018) in blue; arrows indicate breeding colony locations. The 25%, 50%, 75%, and 95% UD levels are shown in progressively lighter shades. BA values represent amount of overlap between each colony in core distribution (50% UD, out of a maximum of 0.5) and overall range (95% UD, out of a maximum of 0.95).

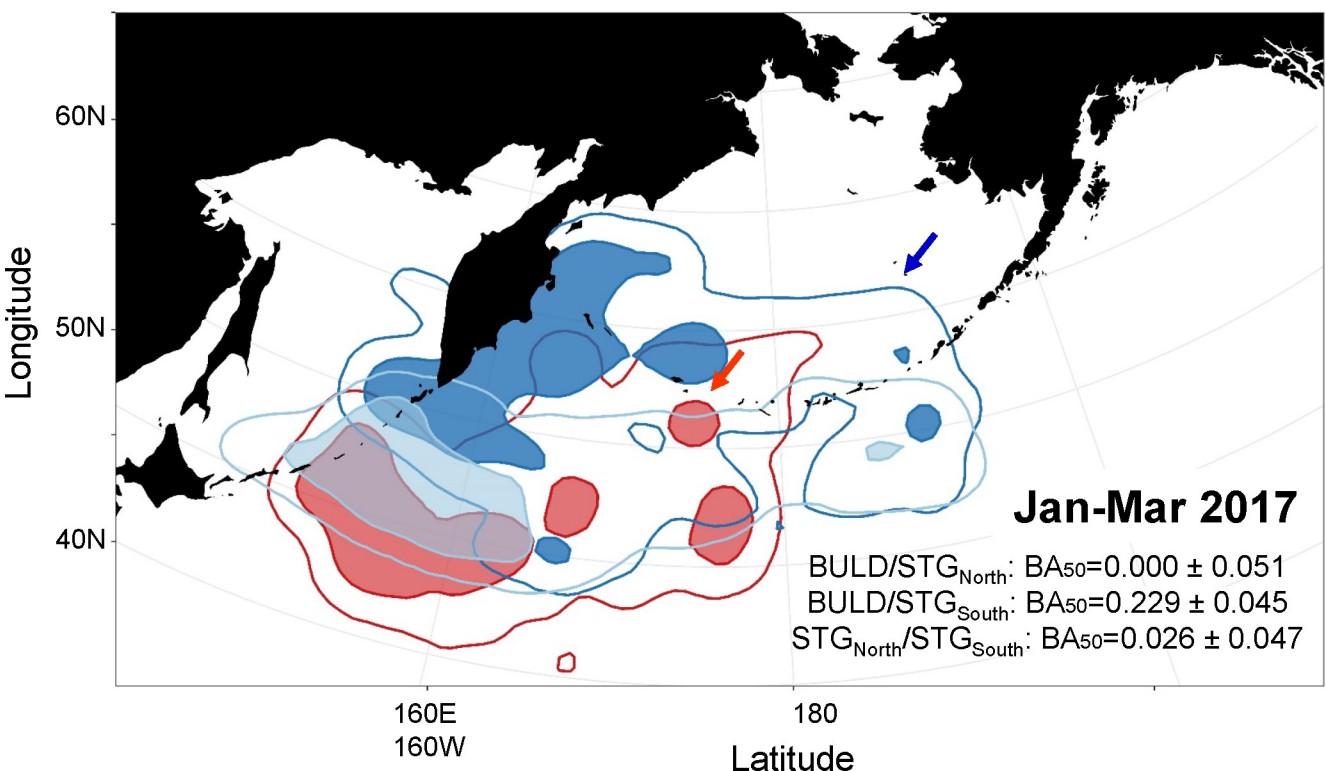

**Fig 4. Overlap of at-sea utilization distributions (UD) of red-legged kittiwakes from Buldir and St. George islands during January-March 2017.** Birds Buldir are shown in red (n = 5), birds from St. George are divided into a northern (n = 11, shown in dark blue) and a southern (n = 17, shown in light blue) group; arrows indicate breeding colony locations. Shaded polygons show 50% UD; empty polygons show 95% UDs. Arrows indicate the locations of breeding colonies at Buldir (red) and St. George (blue). BA values represent amount of overlap (out of a maximum of 0.5) in 50% UDs between the three groups.

little time spent flying (2.1 ± 2.1%) or foraging (3.0 ± 1.4%). Almost all active foraging occurred in the period between an hour before sunrise and sunset, with a large peak in foraging during the hour after sunrise, when birds spent 47.0 ± 6.8% of their time foraging and were on the water very little (6.4 ± 3.8%). Flight occurred primarily during daylight hours, from sunrise to sunset. Kittiwake behavior was largely constant across the non-breeding period, except for a marked increase in the proportion of time birds from both colonies were on the water in January, with a resultant decrease in time flying, during the day and the hour before sunset during that time (Fig 6).

**Table 1. Mean (±SD) distances traveled by red-legged kittiwakes tracked during the non-breeding period (September through April) from Buldir and St. George Islands, Alaska.**

|  | *n* | Cumulative distance (km) | Max. distance from colony (km) |
|---|---|---|---|
| Buldir |  |  |  |
| 2016–17 | 4 | 53,235 ± 1,956 | 2,486 ± 194 |
| 2017–18 | 8 | 53,715 ± 3,691 | 2,657 ± 166 |
| St. George |  |  |  |
| 2016–17 | 27 | 39,192 ± 4,948 | 2,704 ± 290 |
| 2017–18 | 6 | 54,956 ± 3,185 | 3,023 ± 201 |

Values include only birds with complete data records between September and April.

**Table 2. Comparisons of red-legged kittiwake activity patterns between years and colonies.**

| | | Percent time spent: | | | n | James p-value |
|---|---|---|---|---|---|---|
| | | On water | Flying | Foraging | | |
| **Between years** | | | | | | |
| Buldir | 2016–17 | 67.4 ± 3.7% | 19.7 ± 2.4% | 12.9 ± 1.5% | 5 | 0.245 |
| Buldir | 2017–18 | 68.9 ± 3.4% | 19.5 ± 1.8% | 11.7 ± 1.8% | 8 | |
| St. George | 2016–17 | 69.4 ± 2.9% | 18.5 ± 2.4% | 12.0 ± 1.8% | 28 | 0.014 |
| St. George | 2017–18 | 72.6 ± 2.6% | 15.4 ± 2.0% | 12.0 ± 0.8% | 7 | |
| **Between colonies** | | | | | | |
| Buldir | Oct-Mar | 68.5 ± 3.4% | 19.4 ± 1.8% | 12.1 ± 1.7% | 13 | 0.091 |
| St. George | Oct-Mar | 70.1 ± 3.1% | 17.9 ± 2.7% | 12.0 ± 1.6% | 35 | |
| Buldir | Oct-Dec | 68.2 ± 3.8% | 20.1 ± 2.5% | 11.7 ± 1.8% | 13 | 0.114 |
| St. George | Oct-Dec | 66.3 ± 4.2% | 20.7 ± 3.9% | 13.0 ± 1.9% | 35 | |
| Buldir | Jan-Mar | 68.8 ± 3.8% | 18.7 ± 2.1% | 12.5 ± 1.9% | 13 | 0.002 |
| St. George | Jan-Mar | 74.0 ± 3.9% | 15.1 ± 2.7% | 11.0 ± 1.9% | 35 | |

Values include only birds with complete data records between October and March.

## Breeding success

Kittiwakes successfully fledged chicks only at Buldir in 2016; birds experienced complete reproductive failure at Buldir in 2017 and at St. George in both years (Table 3). Most failures occurred early in the breeding period, either during laying or incubation.

## Discussion

Our study simultaneously tracked red-legged kittiwakes from multiple Alaskan breeding colonies over the non-breeding period. In both of the years of the study, birds from the different colonies spent the early winter months in separate regions of the North Pacific, with birds from Buldir concentrated in the southwestern Sea of Okhotsk and birds from St. George mainly occupying the Bering Sea shelf. During late winter in both years, birds from the two colonies converged together along the Kuril-Kamchatka trench and Western Subarctic Gyre before returning to their respective colonies in the spring. Buldir birds spent slightly more time flying and less time sitting on the water during the late winter compared to St. George birds. Otherwise, patterns in activity were similar between both colonies, with birds conducting almost all flying and active foraging during the daytime, including a large peak in foraging concentrated during the hour after sunrise. However, our study was limited by small sample sizes for some years and colonies, and interpretation of any geolocation data should be treated with some caution due to error in location estimates.

Birds from the two colonies exhibited distinctly different strategies during the early winter. Individuals from Buldir traveled thousands of kilometers to their wintering grounds immediately following the breeding season, whereas those from St. George remaining closer to their breeding colony during the early winter months. The post-breeding period can be an important time for birds to replenish their diminished energy reserves after an energetically-costly reproductive effort [e.g., 2, 62–64]. If ample food is available on the Bering Sea shelf close to St. George following the breeding season, red-legged kittiwakes from that colony may be able to replenish their energy reserves without traveling long distances, until later in the winter when lower prey availability pushes birds to migrate elsewhere [39]. Numerous seabird species are abundant across the Bering Sea shelf in the fall and early winter months [65, 66], suggesting prey resources are plentiful in the region at that time. These favorable foraging conditions may

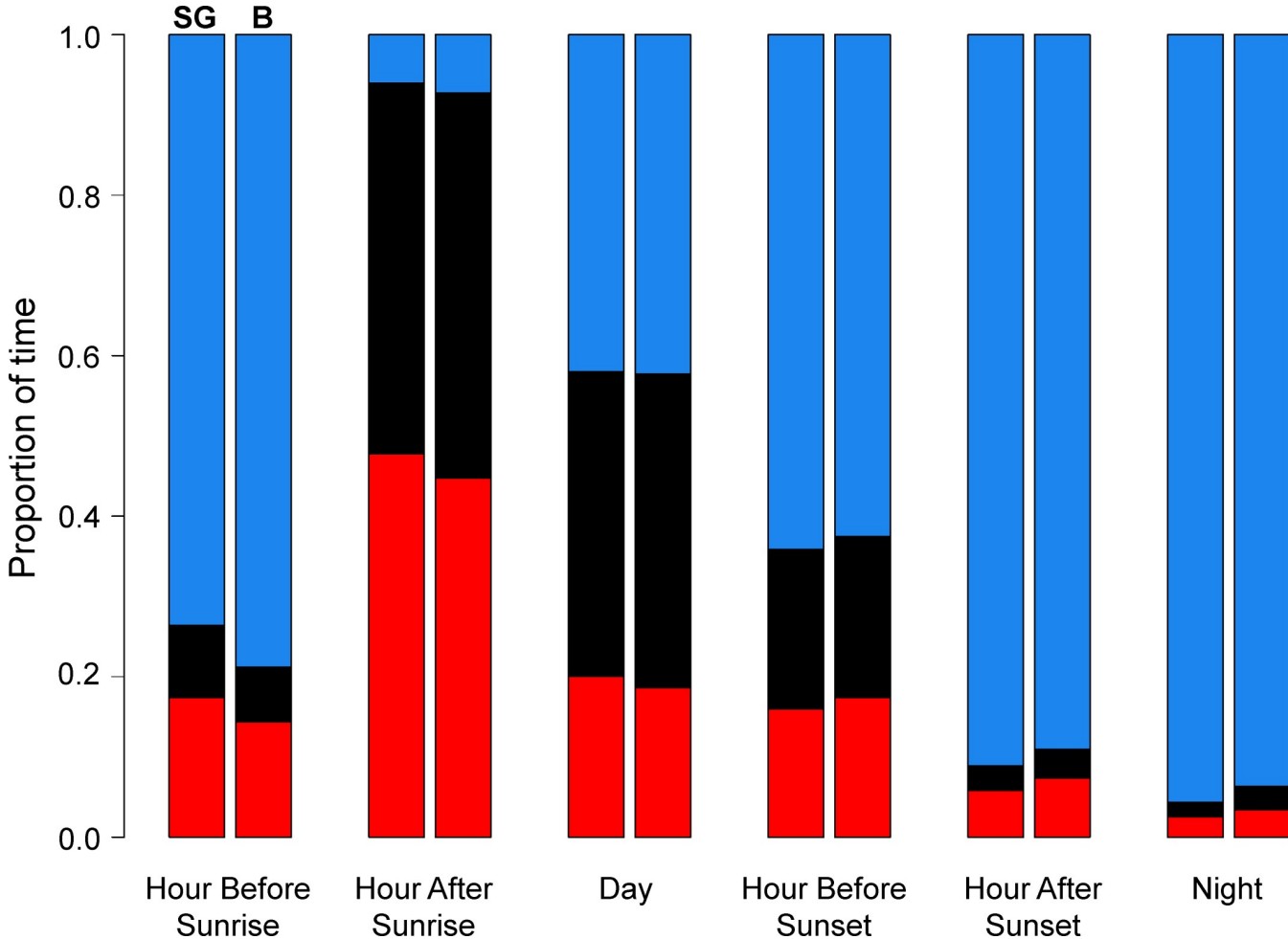

**Fig 5. Proportion of time kittiwakes spent on the water (blue), flying (black), and foraging (red) during different periods of the day from October through March.** For each time of day, birds from St. George (SG; n = 35) are shown in the left column and Buldir (B; n = 13) in the right.

be reliable across years, as red-legged kittiwakes from St. George consistently used the Bering Sea shelf as a post-breeding staging ground in both good and poor reproductive years between 2010–2016, albeit with some interannual variation [39, 49]. In contrast, long distance movements of birds from Buldir immediately post-breeding suggests that food availability around the western Aleutian Islands may less abundant or reliable in the fall and more favorable in the Sea of Okhotsk. Although the passes between islands along the Aleutian Archipelago provide a rich prey supply for seabirds breeding in the Aleutian Islands during the summer months [67], seabird abundance in the western Aleutian Islands declines dramatically during the winter months, signifying prey may be less available [68].

The high degree of overlap in distributions of individuals from Buldir and St. George during the late winter along the Kuril-Kamchatka trench suggests that this is a winter hotspot for red-legged kittiwakes. The waters around and east of the Kuril Islands and in the Western Subarctic Gyre are likely a productive foraging area for kittiwakes, with strong currents and tidal mixing [69, 70] that causes upwelling and eddies to concentrate prey at the ocean surface [71]. Indeed, in previous years, red-legged kittiwakes wintering in the region were found to have low stress levels, indicative of good foraging conditions [39]. St. George and Buldir are the two

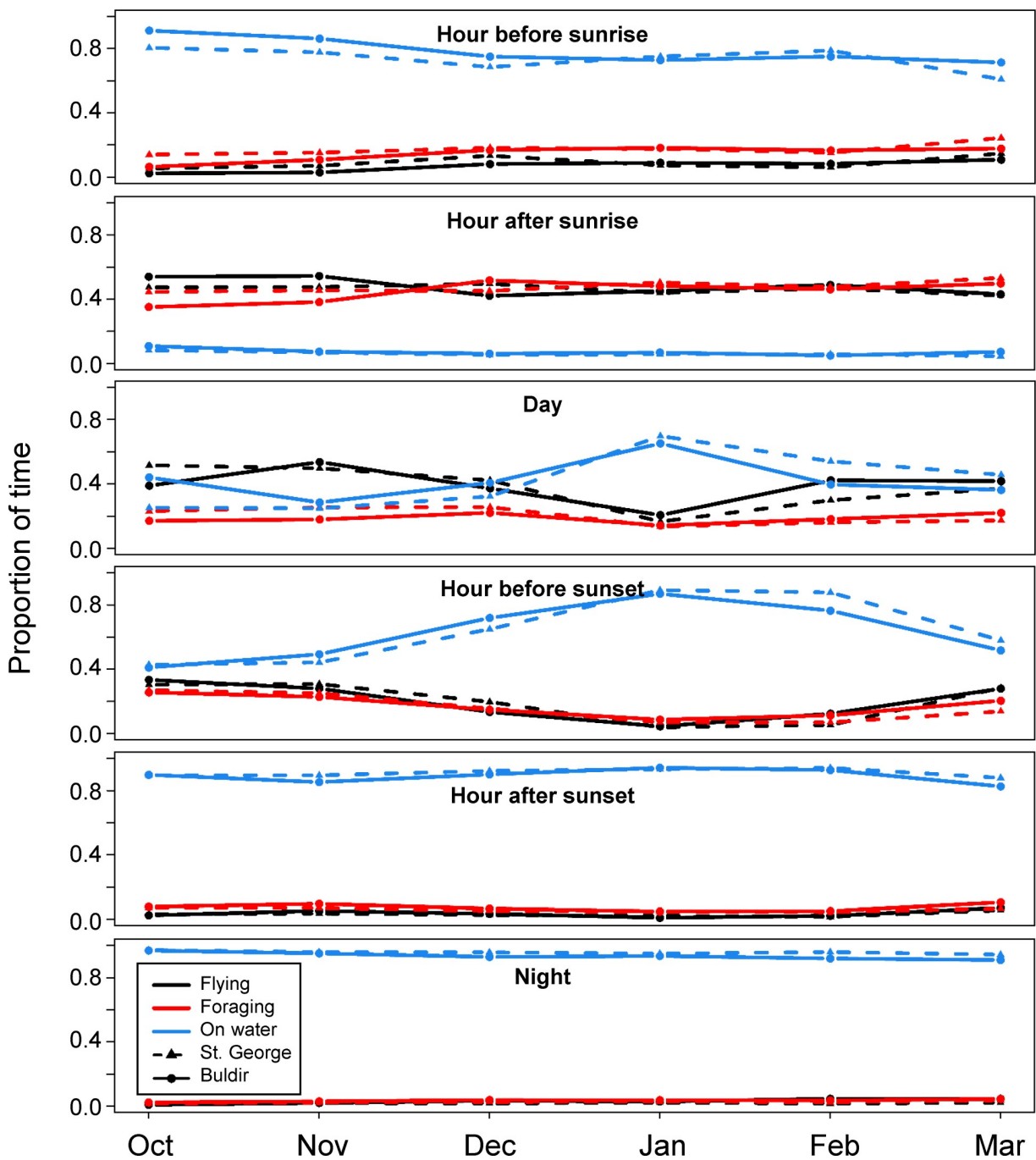

**Fig 6. Proportion of time kittiwakes spent on the water (blue), flying (black), and foraging (red) during different periods of the day in each month across the non-breeding period.** Birds from St. George (n = 35) are shown in triangles and Buldir in circles (n = 13).

largest red-legged kittiwake colonies in Alaska and represent about 85% of the global population [40], so conditions experienced in this shared wintering region could potentially influence a large proportion of the world's red-legged kittiwakes. This wintering region is also in close proximity to the Commander Islands in Russia, the only major red-legged kittiwake colony outside of Alaska (ca 30,000 birds [40]), although migration patterns for that colony are unknown.

**Table 3. Red-legged kittiwake reproductive success at Buldir and St. George Islands, Alaska.**

|  | n | Laying success | Hatching success | Fledging success | Overall breeding success |
|---|---|---|---|---|---|
| Buldir |  |  |  |  |  |
| 2016 | 39 | 0.79 | 0.45 | 0.71 | 0.26 |
| 2017 | 41 | 0.02 | 0.00 | 0.00 | 0.00 |
| St. George |  |  |  |  |  |
| 2016 | 231 | 0.42 | 0.04 | 0.00 | 0.00 |
| 2017 | 153 | 0.08 | 0.00 | 0.00 | 0.00 |

Success represents the proportion of pairs that layed eggs, hatched chicks, and fledged chicks on plots monitored during the breeding season.

Although we tracked kittiwakes for just two winters, it seems likely that these patterns in winter distribution were representative of other years as well. Overall movement patterns and distributions of birds from each colony were fairly consistent during both years of our study, and our data from St. George matched general patterns previously documented during four earlier winters of tracking red-legged kittiwakes at St. George (2010–11, 2013–16 [38]). This fidelity suggests that red-legged kittiwakes may consistently use the same non-breeding areas across years. Furthermore, winter distributions did not appear to vary markedly based on variations in colony-level breeding success. Red-legged kittiwakes from Buldir and St. George did not exhibit different winter movement patterns in years of complete breeding failure (which is an uncommon but not unprecedented event at these colonies [48, 49]) compared to more successful years [38]. However, our study period encompassed two relatively warm winters for the North Pacific region [72, 73] and may not reflect winter movement in colder years. Between 2010 and 2017, winter distributions of red-legged kittiwakes from St. George were most different during the coldest year, when birds stayed farther north on the Bering Shelf for a longer period of time and did not venture as far south along the Kuril Islands as in other warmer years [39]. Therefore, red-legged kittiwakes may use a general overall winter strategy based on persistent foraging resources, but annually make finer scale adjustments based on local conditions.

This study adds to our knowledge of red-legged kittiwake distribution and activity during the non-breeding period. The fact that individuals from separate colonies occupied different areas of the North Pacific during the early winter months but utilized the same wintering area for the late winter could have conservation implications. Birds from each colony may experience different environmental variables earlier in the non-breeding season when geographically segregated, but are likely exposed to similar conditions during late winter on their shared wintering grounds. This late winter overlap along the Kuril-Kamchatka trench and Western Subarctic Gyre highlights the potential importance of this region for the global kittiwake population. Exploring how conditions experienced by red-legged kittiakes during the non-breeding period influence their populations is an important next step to understanding what drives red-legged kittiwake population dynamics. Notably, although we documented timing of winter foraging activity, information on winter diet of red-legged kittiwakes is largely unknown [74]. Data on food availability and diet on the wintering grounds would be a valuable piece to understanding how conditions during the non-breeding period affect this endemic Bering Sea seabird species.

## Acknowledgments

We are grateful to the many field crew members who captured birds at both sites over two summers: T. DeGange, A. Harding, D. Kildaw, C. Kroeger, E. Lefkowitz, F. Mayer, M. Mudge,

K. Srubas, K. Pietrzak, and S. Walden. The R/V *Tila* provided transportation to Buldir Island. We thank Aurore Ponchon and two anonymous reviewers for their helpful reviews of the manuscript. The findings and conclusions in this article are those of the authors and do not necessarily represent the views of USFWS.

## Author Contributions

**Conceptualization:** Brie A. Drummond, Heather M. Renner.

**Data curation:** Rachael A. Orben, Aaron M. Christ, Abram B. Fleishman.

**Formal analysis:** Aaron M. Christ.

**Funding acquisition:** Brie A. Drummond, Rachael A. Orben, Aaron M. Christ, Heather M. Renner, Nora A. Rojek, Marc D. Romano.

**Investigation:** Brie A. Drummond, Rachael A. Orben, Abram B. Fleishman, Heather M. Renner, Nora A. Rojek.

**Methodology:** Brie A. Drummond, Rachael A. Orben, Aaron M. Christ, Abram B. Fleishman.

**Project administration:** Brie A. Drummond.

**Writing – original draft:** Brie A. Drummond, Aaron M. Christ.

**Writing – review & editing:** Brie A. Drummond, Rachael A. Orben, Aaron M. Christ, Abram B. Fleishman, Heather M. Renner, Nora A. Rojek, Marc D. Romano.

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
