## [Decision Letter · Decision Letter 0]

20 Apr 2021

PONE-D-21-08715

Comparing non-breeding distribution and behavior of red-legged kittiwakes from two geographically distant colonies

PLOS ONE

Dear Dr. Drummond,

Thank you for submitting your manuscript to PLOS ONE. After careful consideration, we feel that it has merit but does not fully meet PLOS ONE’s publication criteria as it currently stands. Therefore, we invite you to submit a revised version of the manuscript that addresses the points raised during the review process.

We look forward to receiving your revised manuscript.

Kind regards,

Vitor Hugo Rodrigues Paiva, Ph.D.

Academic Editor

PLOS ONE

Journal Requirements:

We note that one or more of the authors are employed by a commercial company: Conservation Metrics, Inc.

3.1. Please provide an amended Funding Statement declaring this commercial affiliation, as well as a statement regarding the Role of Funders in your study. If the funding organization did not play a role in the study design, data collection and analysis, decision to publish, or preparation of the manuscript and only provided financial support in the form of authors' salaries and/or research materials, please review your statements relating to the author contributions, and ensure you have specifically and accurately indicated the role(s) that these authors had in your study. You can update author roles in the Author Contributions section of the online submission form.

3.2. Please also provide an updated Competing Interests Statement declaring this commercial affiliation along with any other relevant declarations relating to employment, consultancy, patents, products in development, or marketed products, etc.  

6. We note that Figures 1, 3, 4 in your submission contain map images which may be copyrighted. All PLOS content is published under the Creative Commons Attribution License (CC BY 4.0), which means that the manuscript, images, and Supporting Information files will be freely available online, and any third party is permitted to access, download, copy, distribute, and use these materials in any way, even commercially, with proper attribution. For these reasons, we cannot publish previously copyrighted maps or satellite images created using proprietary data, such as Google software (Google Maps, Street View, and Earth). For more information, see our copyright guidelines: http://journals.plos.org/plosone/s/licenses-and-copyright.

6.1.    You may seek permission from the original copyright holder of Figures 1, 3, 4 to publish the content specifically under the CC BY 4.0 license. 

6.2.    If you are unable to obtain permission from the original copyright holder to publish these figures under the CC BY 4.0 license or if the copyright holder’s requirements are incompatible with the CC BY 4.0 license, please either i) remove the figure or ii) supply a replacement figure that complies with the CC BY 4.0 license. Please check copyright information on all replacement figures and update the figure caption with source information. If applicable, please specify in the figure caption text when a figure is similar but not identical to the original image and is therefore for illustrative purposes only.

Reviewers' comments:

Reviewer's Responses to Questions

**Comments to the Author**

1. Is the manuscript technically sound, and do the data support the conclusions?

Reviewer #1: Yes

Reviewer #2: Yes

Reviewer #3: Partly

2. Has the statistical analysis been performed appropriately and rigorously? 

Reviewer #1: Yes

Reviewer #2: Yes

Reviewer #3: Yes

3. Have the authors made all data underlying the findings in their manuscript fully available?

Reviewer #1: Yes

Reviewer #2: Yes

Reviewer #3: Yes

4. Is the manuscript presented in an intelligible fashion and written in standard English?

Reviewer #1: Yes

Reviewer #2: Yes

Reviewer #3: Yes

5. Review Comments to the Author

Reviewer #1: This study documents the non-breeding distribution and activity patterns of red-legged kittiwakes tracked in two different colonies of the Bering Sea. It reveals that individual activity patterns are similar among the two colonies but distributions differ most of the non-breeding season to eventually overlap during late winter.

The manuscript is well-written and clear but I have concerns about the context in which the study is placed and some speculative interpretations.

1) In the abstract and introduction, a lot of emphasis is given on the threats that seabirds could face during the non-breeding season and which may ultimately impact their annual survival. However, the methods applied (individual tracking) cannot provide results directly related to survival and thus, I feel the introduction is not completely appropriate to present the importance and goals of the study.

For example, when reading in the introduction p.4 L.75-77 that adult survival has declined since the 90’s, we have not value and no reference provided. Then, p.4 L.84-87, it is stated that the two study populations are increasing or stable. Therefore, by reading these two contradictory information, it is easy to give more weight to the fact that the study populations are not declining (which is the focus of the manuscript) rather than being worried about the general but unproven annual survival decline of the whole species.

Overall, the introduction sounds dramatic and leads the reader to think that the species is highly threatened, while it doesn’t seem to be the case based on the information given, at least for the study populations.

2) In the introduction, the paragraph stating that seabirds may converge to the same non-breeding area and thus may represent even greater risk for the study populations is biased. Basically, we know that the distributions of seabirds during the non-breeding season is highly variable, with individuals from the same colony wintering in different areas and individuals from different colonies wintering in different areas or in the same areas. All strategies are possible and without previous tracking studies on the study species or closely-related species which may behave the same, it is hard to favour one hypothesis (overlap in winter distribution) over the other (separate distributions). So ok, you have an a priori hypothesis assuming that birds from the 2 colonies may share some wintering areas but given the information available for the hypothesis, I would rather argue that wintering strategies of seabirds are highly variable within and between populations of a same species, that nothing is really known about the winter distribution of red-legged kittiwakes (stable isotopes are only indirect methods) and because of this gap, you want to document it. From the study carried out, your goal is not to study distribution and activity patterns to explain annual survival but rather to compare wintering strategies between two colonies.

So overall, I would encourage to change the introduction into a less dramatic introduction, highlighting that seabird migration is diverse and that you want to know how your species is behaving in winter depending on its breeding colony. And only after presenting the goal of the study, you can present your hypothesis about shared wintering areas. That will definitely improve the introduction. You can discuss the implications of shared overwintering areas for the species but only in the discussion and with a lot of caution.

3) The discussion section about individual condition sounds very speculative. If weight was not measured close to/after chick fledging, it is likely that the weight may not be representative of the final state of individuals at the end of the breeding season. Therefore, I would present first the discussion on the environmental conditions (from p.18 L.367) which are more likely to affect individual distribution, and then, present hypotheses about individual condition but with a lot of caution not to sound too speculative, especially because of the small sample sizes.

Other comments :

Abstract : the abstract is very unbalanced. Based on the previous comments, the introduction is a little bit out of scope and the discussion results are barely presented. I would try to balance a little bit more the abstract to outline its importance and its implications (because the study still brings new valuable knowledge on a seabird species !).

p.2 L 25-26 : First sentence is about adult survival so we expect the paper to be about survival somehow. Nevertheless, the paper never directly relates to it, as individual/population distributions are not real proxies of survival. So the general aim has to be modified.

Second sentence unclear

Methods

p.9 L.198 : replace "species" by « individuals ».

Discussion

p.17 L.336. : Remove the expression « our study was the FIRST ». You don’t need to specify that « you were the first » to stress the importance of your study ! Moreover, it is not the fact that you simultaneously tracked birds from two colonies which has to be highlighted… It is more about the knowledge gap which is filled thanks to your data…

p.19 from L.385 : acknowledge that you still have small sample sizes for some years and colonies and that further studies are needed.

p.20 L.401-403 : this is contradictory to what is stated in the introduction that both colonies are stable or increasing (see also my general comment 1). I would remove the section trying to find some causes of decline in the species (p.20 403-421), it is way too speculative based on the given evidence in the manuscript that the species is declining.

p.21 L.428-429 : I don’t agree with this sentence. Kittiwakes have the same activity patterns, meaning they need the same amount of time spent foraging and thus have the same energy requirements. But it doesn’t mean that they feed on the same prey species.

Conclusion: I miss a last paragraph summarizing the take-home message and the wider implications of the study.

Reviewer #2: This is an interesting descriptive paper that compares winter distributions in two colonies of a very interesting and unusual small gull, the red-legged kittiwake. The field and analytical methods are sound and the results have important conservation implications. The paper is generally well written, and of appropriate length. There are quite a few Figures, but most or all are warranted. I have a few concerns, related to small sample sizes, especially at Buldir, although the results are probably robust. I also thought there was a bit too much speculation in the Discussion, which should focus less on the weaker components of the paper (activity patterns, body condition), and more on the strengths of the paper (distribution.)

A few specific comments:

Introduction

Line 53: Could maybe replace “are crucial to” with “can have marked effects on” or something similar

Line 56: Given most seabirds spend all of winter at sea, you could delete “when the majority of time is spent at sea”

Line 58-61: I might reword to say “widely distributed, habitat generalists may be more resilient to environmental change than more narrowly distributed habitat specialists” or something like that.

Line 64: Could change to “during the winters of 2007 to 2010”

Line 69: Could change to “colonial, cliff-nesting” to provide more information.

Line 76: Could name the two colonies here.

Line 83: Tell us what years you tracked the birds here.

Line 91 to 93: I like the specific predictions, but you should justify why you predicted that behavior would be invariant among individuals.

Methods

Line 106-107: From above, below, or both?

Line 114: Perhaps report on the range in masses of tagged birds here?

Line 129-131: What is the clutch size in these birds?

Line 157-158: No need to repeat the sample sizes here

Line 182-183: Because sample sizes for the different components of the study vary a bit, you might consider putting the information in a Table to avoid repeatedly breaking up the text.

Line 183: “a bird that spends”. There are other occasions in the paper as well.

Line 192-200: Use of these ‘condition indices” derived by regressing mass on linear morphometrics has received harsh criticism, for good reason. For these birds, do males and females differ in size? And would mass not vary considerably depending (for example) on how long the bird had spent on the nest prior to capture, and the time in the season when it was captured?

Results

Fig. 2, Lines 237-238: You might spell this out in more detail, so that the Figure plus caption can stand on its own, separate from the rest of the paper.

Line 248: Should be “was higher”

Table 3: It looks like almost no birds laid eggs in 2017 at either colony, and none hatched eggs. Numbers in 2016 at St George were also very low. So would a valid intepretation of these results be that breeding birds actively maintain higher mass than non-breeding birds? I sense that your interpretation is that causation works in the other direction, i.e., that birds fail because they are light in mass = in poor condition.

Discussion

Line 354-366: This paragraph encapsulates the issue with condition indices, and their intepretation. This frankly comes off as a lot of arm-waving, and in my opinion, is not a strength of the paper. I would eliminate these sections related to condition unless there is a more clear and compelling interpretation available.

A fair bit of the material in the Discussion is speculative, especially related to seasonal and colony differences in condition and activity patterns. Again, I do not think these are the strengths of the paper. I would probably try to keep the Discussion focused more on the distributions – where, when and (as much as possible), why.

Would it be possible to make at least brief comparisons with other non-Larus, small, northern gulls like Sabine’s, Ross’s, and Ivory? I think those species have all been tracked with GLS tags now.

Reviewer #3: Overall, this study provides a well written description of the broad behaviour and non-breeding season distribution of red-legged kittiwakes from two breeding colonies. The study focuses on a globally important population and therefore has a broad interest to the scientific field. The work also manages to culminate in several concise and interesting conclusions.

While the study has two consecutive years of data, the sample sizes across years is relatively small. The authors do a good job in acknowledging this and none of the conclusions made by the study overstep the bounds of the limited sample size. The study is also fortunate in that there has been work done in the area previously, to help put their results into perspective.

The methods and analysis used are relatively typical in this field and I therefore have no concerns.

I am happy to make a recommendation for publication of this paper, but I have one major comment on the final paragraph (starting line 428) of the discussion, which I discuss below and believe needs to be addressed.

The data obtained from the immersion sensor on the GLS tags is not enough to reach the conclusion that individuals from both colonies are “feeding on similar prey items throughout the winter period”. Additionally, the activity around sunrise and sunset alone is not enough to conclude that the birds are “mainly consuming diel vertically migrating prey”, as foraging activity during these periods does not preclude the consumption of other prey types, without evidence to verify the prey consumed. Finally, the comment that this study adds evidence to the birds being specialist foragers is far too speculative given the data and results from this study. I would suggest rewriting this paragraph to focus on the results obtained in this study, mainly: diel foraging patterns and other behavioral differences and similarities, while avoiding making speculative points on wider foraging behaviour and prey consumption.

Other than these broad comments, I have added a few minor and specific points below:

A comment on the degree of accuracy of geolocators could be helpful, along with a comment in the discussion mentioning how the error in location estimates could also affect the degree of overlap between colonies and the distance travelled. As is stated in the paper, probGLS should minimize these errors and they are very unlikely to affect your results, but I think it is important to mention and quantify (using a value from the literature) them regardless.

Line 27: Suggest you change “widespread impact on the population” to “widespread impact on multiple breeding populations”, as this better conveys your point that changes in a multi-colony non-breeding ground site could have repercussions on multiple breeding populations.

Line 74&75: There is an open “[“ which ends with a “)”. Suggest a change of the “)” to a “]”

Line 97: Include the dimensions of the tag.

Line 104: Include the units for the value of conductivity.

Line 106: It is worth noting in this paragraph whether birds were retagged or not i.e. “Birds were not retagged upon retrieval of GLS devices”

Line 144&145: This sentence is not immediately clear what the “respectively” refers to, as there are three values and two overlaps. I suggest a slight rewording of this sentence just to improve clarity, as this is an important point which puts a lot of the results in context.

Line 189: Include the citation for twGeos (ref 50)

Line 347: In this paragraph it would be nice to know how often complete colony failure happens. Is this a rare event or a relatively common occurrence?

Lines 358 – 360: A reference is needed here which demonstrates the birds have flexibility in molt timings and that timing can be driven by reproductive success, otherwise this point is too speculative.

Lines 395-398: A good point, but the sentence reads a bit awkwardly at the moment, perhaps reword slightly.

Line 404-405: A reference linking food reliability to survival is needed here.

Line 424: Need a reference to the fact the colonies represent 85% of the global population.

Figure 4: Showing the BA50 between the north and south sides of St. George colony on this map would be helpful

6. PLOS authors have the option to publish the peer review history of their article (what does this mean?). If published, this will include your full peer review and any attached files.

Reviewer #1: **Yes: **Dr Aurore Ponchon

Reviewer #2: No

Reviewer #3: No

---

## [Author Response · Author response to Decision Letter 0]

4 Jun 2021

PONE-D-21-08715 Response to Reviewers

Please see response (noted with asterisks) to journal requirements and reviewers below:

Journal Requirements:

*We have done so.

*We have updated the Funding Information (see online submission form) and Funding Statement (see cover letter) so that they match.

We note that one or more of the authors are employed by a commercial company: Conservation Metrics, Inc.

3.1. Please provide an amended Funding Statement declaring this commercial affiliation, as well as a statement regarding the Role of Funders in your study. If the funding organization did not play a role in the study design, data collection and analysis, decision to publish, or preparation of the manuscript and only provided financial support in the form of authors' salaries and/or research materials, please review your statements relating to the author contributions, and ensure you have specifically and accurately indicated the role(s) that these authors had in your study. You can update author roles in the Author Contributions section of the online submission form.

*We have updated the author roles in the online submission form.

*We have updated our Funding Statement accordingly (see cover letter).

3.2. Please also provide an updated Competing Interests Statement declaring this commercial affiliation along with any other relevant declarations relating to employment, consultancy, patents, products in development, or marketed products, etc. 

*We have updated our Competing Interests Statement accordingly (see cover letter).

*We have provided repository information in an updated Data Availability statement (see cover letter).

*Done (Lines 70-73).

6. We note that Figures 1, 3, 4 in your submission contain map images which may be copyrighted. All PLOS content is published under the Creative Commons Attribution License (CC BY 4.0), which means that the manuscript, images, and Supporting Information files will be freely available online, and any third party is permitted to access, download, copy, distribute, and use these materials in any way, even commercially, with proper attribution. For these reasons, we cannot publish previously copyrighted maps or satellite images created using proprietary data, such as Google software (Google Maps, Street View, and Earth). For more information, see our copyright guidelines: http://journals.plos.org/plosone/s/licenses-and-copyright.

6.1. You may seek permission from the original copyright holder of Figures 1, 3, 4 to publish the content specifically under the CC BY 4.0 license. 

6.2. If you are unable to obtain permission from the original copyright holder to publish these figures under the CC BY 4.0 license or if the copyright holder’s requirements are incompatible with the CC BY 4.0 license, please either i) remove the figure or ii) supply a replacement figure that complies with the CC BY 4.0 license. Please check copyright information on all replacement figures and update the figure caption with source information. If applicable, please specify in the figure caption text when a figure is similar but not identical to the original image and is therefore for illustrative purposes only.

*Our maps were created using an open source R package called mapdata package 2.3.0 (Brownrigg R. Mapdata: Extra Map Databases. Version 2.3.0 [R package]. 2018. Available at: https://CRAN.R-project.org/package=mapdata), and so are not proprietary data and can be published under a CC BY 4.0 license. We have noted this in the manuscript (Lines 175-176) and added a citation to the package in the manuscript (Lines 537-538).

 

Reviewer #1: This study documents the non-breeding distribution and activity patterns of red-legged kittiwakes tracked in two different colonies of the Bering Sea. It reveals that individual activity patterns are similar among the two colonies but distributions differ most of the non-breeding season to eventually overlap during late winter.

The manuscript is well-written and clear but I have concerns about the context in which the study is placed and some speculative interpretations.

1) In the abstract and introduction, a lot of emphasis is given on the threats that seabirds could face during the non-breeding season and which may ultimately impact their annual survival. However, the methods applied (individual tracking) cannot provide results directly related to survival and thus, I feel the introduction is not completely appropriate to present the importance and goals of the study.

For example, when reading in the introduction p.4 L.75-77 that adult survival has declined since the 90’s, we have not value and no reference provided. Then, p.4 L.84-87, it is stated that the two study populations are increasing or stable. Therefore, by reading these two contradictory information, it is easy to give more weight to the fact that the study populations are not declining (which is the focus of the manuscript) rather than being worried about the general but unproven annual survival decline of the whole species.

Overall, the introduction sounds dramatic and leads the reader to think that the species is highly threatened, while it doesn’t seem to be the case based on the information given, at least for the study populations.

*We have rewritten parts of the Abstract and Introduction to remove any emphasis on adult survival and instead highlight the distribution aspects of the study. 

2) In the introduction, the paragraph stating that seabirds may converge to the same non-breeding area and thus may represent even greater risk for the study populations is biased. Basically, we know that the distributions of seabirds during the non-breeding season is highly variable, with individuals from the same colony wintering in different areas and individuals from different colonies wintering in different areas or in the same areas. All strategies are possible and without previous tracking studies on the study species or closely-related species which may behave the same, it is hard to favour one hypothesis (overlap in winter distribution) over the other (separate distributions). So ok, you have an a priori hypothesis assuming that birds from the 2 colonies may share some wintering areas but given the information available for the hypothesis, I would rather argue that wintering strategies of seabirds are highly variable within and between populations of a same species, that nothing is really known about the winter distribution of red-legged kittiwakes (stable isotopes are only indirect methods) and because of this gap, you want to document it. From the study carried out, your goal is not to study distribution and activity patterns to explain annual survival but rather to compare wintering strategies between two colonies.

So overall, I would encourage to change the introduction into a less dramatic introduction, highlighting that seabird migration is diverse and that you want to know how your species is behaving in winter depending on its breeding colony. And only after presenting the goal of the study, you can present your hypothesis about shared wintering areas. That will definitely improve the introduction. You can discuss the implications of shared overwintering areas for the species but only in the discussion and with a lot of caution. 

*While it is true that seabird populations can have different wintering strategies, it is incorrect that nothing is really known about the winter distribution of red-legged kittiwakes. Prior to this study, red-legged kittiwakes were tracked from the St. George colony over four winters, during which time their distributions were highly similar across years. We formed the hypothesis that birds from both St. George and Buldir colonies would overlap based on that earlier work, as the consistent areas used by St. George birds in previous winters were geographically close to the Buldir colony. We have emphasize the existence of this previous work (Lines 52-53) and clarified it as the basis for our initial hypotheses in the Introduction (Lines 61-64).

3) The discussion section about individual condition sounds very speculative. If weight was not measured close to/after chick fledging, it is likely that the weight may not be representative of the final state of individuals at the end of the breeding season. Therefore, I would present first the discussion on the environmental conditions (from p.18 L.367) which are more likely to affect individual distribution, and then, present hypotheses about individual condition but with a lot of caution not to sound too speculative, especially because of the small sample sizes.

*We acknowledge the problems with the condition index dataset and have removed it entirely from the manuscript.

Other comments :

Abstract : the abstract is very unbalanced. Based on the previous comments, the introduction is a little bit out of scope and the discussion results are barely presented. I would try to balance a little bit more the abstract to outline its importance and its implications (because the study still brings new valuable knowledge on a seabird species !).

*We have rewritten parts of the Abstract to change the scope away from survival, and emphasize some more of the major implications of the work.

p.2 L 25-26 : First sentence is about adult survival so we expect the paper to be about survival somehow. Nevertheless, the paper never directly relates to it, as individual/population distributions are not real proxies of survival. So the general aim has to be modified.

*We have rewritten this sentence to not concentrate on survival.

Second sentence unclear

*We have attempted to clarify this.

Methods

p.9 L.198 : replace "species" by « individuals ».

*Done.

Discussion

p.17 L.336. : Remove the expression « our study was the FIRST ». You don’t need to specify that « you were the first » to stress the importance of your study ! Moreover, it is not the fact that you simultaneously tracked birds from two colonies which has to be highlighted… It is more about the knowledge gap which is filled thanks to your data…

*Done.

p.19 from L.385 : acknowledge that you still have small sample sizes for some years and colonies and that further studies are needed.

*Added earlier in the Discussion (Line 306).

p.20 L.401-403 : this is contradictory to what is stated in the introduction that both colonies are stable or increasing (see also my general comment 1). I would remove the section trying to find some causes of decline in the species (p.20 403-421), it is way too speculative based on the given evidence in the manuscript that the species is declining.

*We have removed the reference to survival, to avoid speculation on what is causing declines. While we still discuss some aspects of conditions in the region, due to its apparent importance as a hotspot during the winter months, we have eliminated much of this section and removed any attempts to link such conditions to kittiwake population dynamics.

p.21 L.428-429 : I don’t agree with this sentence. Kittiwakes have the same activity patterns, meaning they need the same amount of time spent foraging and thus have the same energy requirements. But it doesn’t mean that they feed on the same prey species.

*We have removed this.

Conclusion: I miss a last paragraph summarizing the take-home message and the wider implications of the study.

*We have added a concluding paragraph to the Discussion (Lines 357-370).

 

Reviewer #2: This is an interesting descriptive paper that compares winter distributions in two colonies of a very interesting and unusual small gull, the red-legged kittiwake. The field and analytical methods are sound and the results have important conservation implications. The paper is generally well written, and of appropriate length. There are quite a few Figures, but most or all are warranted. I have a few concerns, related to small sample sizes, especially at Buldir, although the results are probably robust. I also thought there was a bit too much speculation in the Discussion, which should focus less on the weaker components of the paper (activity patterns, body condition), and more on the strengths of the paper (distribution.)

*We have removed some of the more speculative sections from the Discussion, including activity patterns and body condition.

A few specific comments:

Introduction

Line 53: Could maybe replace “are crucial to” with “can have marked effects on” or something similar

*Done.

Line 56: Given most seabirds spend all of winter at sea, you could delete “when the majority of time is spent at sea”

*Done.

Line 58-61: I might reword to say “widely distributed, habitat generalists may be more resilient to environmental change than more narrowly distributed habitat specialists” or something like that.

*Done.

Line 64: Could change to “during the winters of 2007 to 2010”

*Done.

Line 69: Could change to “colonial, cliff-nesting” to provide more information.

*Done.

Line 76: Could name the two colonies here.

*This sentence has been removed.

Line 83: Tell us what years you tracked the birds here.

*Done.

Line 91 to 93: I like the specific predictions, but you should justify why you predicted that behavior would be invariant among individuals.

*Done (Lines 66-67).

Methods

Line 106-107: From above, below, or both?

*This detail was added (Line 86).

Line 114: Perhaps report on the range in masses of tagged birds here?

*This detail was added (Line 93).

Line 129-131: What is the clutch size in these birds?

*This detail was added (Line 168).

Line 157-158: No need to repeat the sample sizes here

*Deleted.

Line 182-183: Because sample sizes for the different components of the study vary a bit, you might consider putting the information in a Table to avoid repeatedly breaking up the text.

*We chose not to add an additional table for sample sizes because sample size information is already presented in Tables 1-2 in the Results.

Line 183: “a bird that spends”. There are other occasions in the paper as well.

*These were corrected.

Line 192-200: Use of these ‘condition indices” derived by regressing mass on linear morphometrics has received harsh criticism, for good reason. For these birds, do males and females differ in size? And would mass not vary considerably depending (for example) on how long the bird had spent on the nest prior to capture, and the time in the season when it was captured?

*We acknowledge the problems with our condition index and have chosen to remove that dataset from the manuscript.

Results

Fig. 2, Lines 237-238: You might spell this out in more detail, so that the Figure plus caption can stand on its own, separate from the rest of the paper.

*More details are included in the Figure legend (Lines 202-206).

Line 248: Should be “was higher”

*This was corrected.

Table 3: It looks like almost no birds laid eggs in 2017 at either colony, and none hatched eggs. Numbers in 2016 at St George were also very low. So would a valid intepretation of these results be that breeding birds actively maintain higher mass than non-breeding birds? I sense that your interpretation is that causation works in the other direction, i.e., that birds fail because they are light in mass = in poor condition.

*We have removed the condition dataset from the manuscript.

Discussion

Line 354-366: This paragraph encapsulates the issue with condition indices, and their intepretation. This frankly comes off as a lot of arm-waving, and in my opinion, is not a strength of the paper. I would eliminate these sections related to condition unless there is a more clear and compelling interpretation available.

*We agree and have removed the condition discussion from the manuscript.

A fair bit of the material in the Discussion is speculative, especially related to seasonal and colony differences in condition and activity patterns. Again, I do not think these are the strengths of the paper. I would probably try to keep the Discussion focused more on the distributions – where, when and (as much as possible), why.

*We have reduced the more speculative parts of the discussion, especially in removing sections discussing seasonal and colony differences in condition and activity patterns.

Would it be possible to make at least brief comparisons with other non-Larus, small, northern gulls like Sabine’s, Ross’s, and Ivory? I think those species have all been tracked with GLS tags now.

*While some data for those species exist, we did not think comparisons with these species fit into the scope of our manuscript, as they have very different breeding areas and do not winter in the North Pacific.

 

Reviewer #3: Overall, this study provides a well written description of the broad behaviour and non-breeding season distribution of red-legged kittiwakes from two breeding colonies. The study focuses on a globally important population and therefore has a broad interest to the scientific field. The work also manages to culminate in several concise and interesting conclusions.

While the study has two consecutive years of data, the sample sizes across years is relatively small. The authors do a good job in acknowledging this and none of the conclusions made by the study overstep the bounds of the limited sample size. The study is also fortunate in that there has been work done in the area previously, to help put their results into perspective.

The methods and analysis used are relatively typical in this field and I therefore have no concerns.

I am happy to make a recommendation for publication of this paper, but I have one major comment on the final paragraph (starting line 428) of the discussion, which I discuss below and believe needs to be addressed.

The data obtained from the immersion sensor on the GLS tags is not enough to reach the conclusion that individuals from both colonies are “feeding on similar prey items throughout the winter period”. Additionally, the activity around sunrise and sunset alone is not enough to conclude that the birds are “mainly consuming diel vertically migrating prey”, as foraging activity during these periods does not preclude the consumption of other prey types, without evidence to verify the prey consumed. Finally, the comment that this study adds evidence to the birds being specialist foragers is far too speculative given the data and results from this study. I would suggest rewriting this paragraph to focus on the results obtained in this study, mainly: diel foraging patterns and other behavioral differences and similarities, while avoiding making speculative points on wider foraging behaviour and prey consumption.

*We have removed this section from the Discussion. The results from the feeding activity is reported earlier in the Discussion (Lines 305-306) but without any of the more speculative discussion points.

Other than these broad comments, I have added a few minor and specific points below:

A comment on the degree of accuracy of geolocators could be helpful, along with a comment in the discussion mentioning how the error in location estimates could also affect the degree of overlap between colonies and the distance travelled. As is stated in the paper, probGLS should minimize these errors and they are very unlikely to affect your results, but I think it is important to mention and quantify (using a value from the literature) them regardless.

*We have added details about the error in location estimates in the methods (Lines 115-120) and mentioned them in the discussion (Lines 307-308). 

Line 27: Suggest you change “widespread impact on the population” to “widespread impact on multiple breeding populations”, as this better conveys your point that changes in a multi-colony non-breeding ground site could have repercussions on multiple breeding populations.

*Done.

Line 74&75: There is an open “[“ which ends with a “)”. Suggest a change of the “)” to a “]”

*Corrected.

Line 97: Include the dimensions of the tag.

*This detail was added (Line 92).

Line 104: Include the units for the value of conductivity.

*There are no units for the value of conductivity to report. The geolocation loggers we used measure conductivity on an arbitrary 0–127 scale, in which a value of 63 is recommended as the cutoﬀ between brackish and fresh water (according to Migrate Technology, the company who makes the geolocation loggers). To clarify that this isn’t an exact measurement, we changed “value” to “score” (Line 83).

Line 106: It is worth noting in this paragraph whether birds were retagged or not i.e. “Birds were not retagged upon retrieval of GLS devices”

*Added (Line 89).

Line 144&145: This sentence is not immediately clear what the “respectively” refers to, as there are three values and two overlaps. I suggest a slight rewording of this sentence just to improve clarity, as this is an important point which puts a lot of the results in context.

*We reworded this for clarity.

Line 189: Include the citation for twGeos (ref 50)

*Done.

Line 347: In this paragraph it would be nice to know how often complete colony failure happens. Is this a rare event or a relatively common occurrence?

*We have added this information (Lines 348-349).

Lines 358 – 360: A reference is needed here which demonstrates the birds have flexibility in molt timings and that timing can be driven by reproductive success, otherwise this point is too speculative.

*This sections has been removed.

Lines 395-398: A good point, but the sentence reads a bit awkwardly at the moment, perhaps reword slightly.

*We have reworded this sentence.

Line 404-405: A reference linking food reliability to survival is needed here.

*This section has been removed.

Line 424: Need a reference to the fact the colonies represent 85% of the global population.

*Done.

Figure 4: Showing the BA50 between the north and south sides of St. George colony on this map would be helpful

*We have added this.

---

## [Decision Letter · Decision Letter 1]

2 Jul 2021

Comparing non-breeding distribution and behavior of red-legged kittiwakes from two geographically distant colonies

PONE-D-21-08715R1

Dear Dr. Drummond,

We’re pleased to inform you that your manuscript has been judged scientifically suitable for publication and will be formally accepted for publication once it meets all outstanding technical requirements.

Kind regards,

Vitor Hugo Rodrigues Paiva, Ph.D.

Academic Editor

PLOS ONE

Additional Editor Comments (optional):

Reviewers' comments:

Reviewer's Responses to Questions

**Comments to the Author**

1. If the authors have adequately addressed your comments raised in a previous round of review and you feel that this manuscript is now acceptable for publication, you may indicate that here to bypass the “Comments to the Author” section, enter your conflict of interest statement in the “Confidential to Editor” section, and submit your "Accept" recommendation.

Reviewer #1: All comments have been addressed

Reviewer #3: All comments have been addressed

2. Is the manuscript technically sound, and do the data support the conclusions?

Reviewer #1: Yes

Reviewer #3: Yes

3. Has the statistical analysis been performed appropriately and rigorously? 

Reviewer #1: Yes

Reviewer #3: Yes

4. Have the authors made all data underlying the findings in their manuscript fully available?

Reviewer #1: Yes

Reviewer #3: Yes

5. Is the manuscript presented in an intelligible fashion and written in standard English?

Reviewer #1: Yes

Reviewer #3: Yes

6. Review Comments to the Author

Reviewer #1: I am pleased to see that the authors have dealt satisfactorily with the criticisms that other reviewers and I had raised regarding the original version of this manuscript dealing with the winter migration of red-legged kittiwakes breeding in two different colonies of the Bering Sea. The revised version has been substantially improved, notably by the more appropriate context the study is replaced in and the less speculative discussion describing more closely the results of the study. The manuscript is easier to read and we now understand the importance of this work in the right context.

Reviewer #3: (No Response)

7. PLOS authors have the option to publish the peer review history of their article (what does this mean?). If published, this will include your full peer review and any attached files.

Reviewer #1: **Yes: **Dr Aurore Ponchon

Reviewer #3: No

---

## [Editor Report · Acceptance letter]

8 Jul 2021

PONE-D-21-08715R1 

Comparing non-breeding distribution and behavior of red-legged kittiwakes from two geographically distant colonies 

Dear Dr. Drummond:

I'm pleased to inform you that your manuscript has been deemed suitable for publication in PLOS ONE. Congratulations! Your manuscript is now with our production department. 

Kind regards, 

on behalf of

Dr. Vitor Hugo Rodrigues Paiva 

Academic Editor

PLOS ONE